# Emotional and Behavioral Problems Among 3- to 5-Year-Olds Left-Behind Children in Poor Rural Areas of Hunan Province: A Cross-Sectional Study

**DOI:** 10.3390/ijerph16214188

**Published:** 2019-10-29

**Authors:** Jing Luo, Jiaojiao Zou, Meimei Ji, Tong Yuan, Mei Sun, Qian Lin

**Affiliations:** 1Department of Nutrition Science and Food Hygiene, Xiangya School of Public Health, Central South University, 110 Xiangya Road, Changsha 410078, Hunan, China; luojing2546@csu.edu.cn (J.L.); zjj227@csu.edu.cn (J.Z.); jimeimei@csu.edu.cn (M.J.); yuantong@csu.edu.cn (T.Y.); 2School of Nursing Central South University, 172 Tongzipo Road, Changsha 410013, Hunan, China

**Keywords:** left-behind children, preschool left-behind children, emotional and behavioral problems, caregiver, rural area, China

## Abstract

The term left-behind children (LBC) refers to underage children who have been separated from their father/mother or parents for a long period of time. At present, there are few studies on the emotional and behavioral problems of three- to five-year-old LBC in poor rural areas of China. In this study, random cluster sampling was used to recruit caregivers of left-behind children (CLBC) aged three to five years in two poor rural areas in Hunan Province. General demographic data of LBC and their caregivers were collected via face-to-face questionnaires. Children’s emotional and behavioral problems were assessed by the Strength and Difficulties Questionnaire (SDQ). Among 557 LBC, the prevalence of a total difficulties score in an abnormal/borderline range was 27.6–50.6%. The most common problem of LBC was hyperactivity, with a rate of 33.6%. Compared with boys, girls had more emotional problems (*p* < 0.05) and fewer hyperactivity disorders (*p* < 0.01). Factors related to the emotional and behavioral problems of LBC were the LBC’s age, the number of sick people at home, and the CLBC’s willingness to take care of the LBC. The detection rate of emotional and behavioral problems of three- to five-year-old LBC in poor rural areas is higher than that of children of the same age in urban areas and Western developed countries. There were gender differences in hyperactivity and emotional symptoms. Poor care will increase the risk of children’s emotional and behavioral abnormalities.

## 1. Introduction

In recent decades, China’s economy has been developing rapidly but the distribution of public resources, such as education, health care, and employment, is still uneven between urban and rural areas, especially among the special group of left-behind children (LBC) in rural areas. Due to China’s urbanization process, the number of young and middle-aged members of the rural labor force moving to cities has increased, and the number of LBC has thus also increased. According to the national population sampling survey conducted in 2015, the number of LBC in rural areas in China is 68,766 million, among whom 40.34% are LBC under the age of five [1]. However, most preschool-aged LBC in China are scattered in poor rural areas without LBC care institutions, such as kindergartens, making it difficult for the country’s existing basic public health service policies and related mental health interventions to benefit this vulnerable group. This study also found that the utilization of health services by LBC and their caregivers in poor rural areas was extremely low [2,3].

To our knowledge, children who live with their parents have better physical and mental health than children who live with their grandparents or single parents [4,5]. Parents play an important role in the establishment of a child’s secure attachment relationships, emotional support, self-regulation, social skills development, and practical assistance [6]. Therefore, children that have only grandparents or single parents as caregivers are more likely to have externalized behavioral problems and internalized emotional problems than those whose caregivers are both parents [4,5]. In fact, previous studies have shown that one or both parents’ death, divorce or remarriage, and long-term separation from children can have a negative impact on a child’s emotional and behavioral functions [7,8,9,10,11]. Not surprisingly, LBC in rural areas in China have been separated from their parents for a long time and have been monitored by relatives or neighbors, which increases the risk of emotional and behavioral problems [12,13]. Some studies have also reported that LBC are more prone to depressive symptoms and behavioral problems than non-LBC [14,15]. In addition, previous literature has reported that LBC have a higher risk of emotional and behavioral problems when they are separated from their parents at a younger age and for a longer time [12,16,17]. Considering that there are 68 million LBC in China and over 40% of them are LBC under the age of five [1], it is advisable to pay attention to their emotional and behavioral problems.

The emotional and behavioral problems of children are currently an area of great concern [18]. As early as 1993, the American Academy of Pediatrics adopted a policy statement on “new diseases”, which introduced “new morbidity” in pediatric practice. These “new diseases” mainly include school-related problems (such as learning disabilities and attention difficulties), emotional and anxiety disorders in children and adolescents, as well as adolescent alcoholism, violence, drug abuse, suicide, etc. [19]. A recent national report from the American Academy of Pediatrics shows that the prevalence of emotional/behavioral disorders in children is 11% to 20% for all ages [20]. The emotional and behavioral disorders of children have become main chronic pediatric diseases affecting their psychosocial function. In recent years, American children’s disability caused by mental health problems has surpassed physical diseases, becoming one of the top five diseases of children’s disability [21]. Teekavanich et al. reviewed previous studies and found that the prevalence of emotional and behavioral problems among preschool children ranged from 7% to 25% [22]. If emotional and behavioral problems in preschool- or school-aged children are not found and treated as early as possible, these individuals may have emotional and behavioral disorders in adulthood. These disorders mainly include the withdrawal of internalized symptoms, anxiety, depression, and hyperactivity disorder with externalized symptoms, if serious. In the case of behavioral disorders, it is likely that they will eventually develop into anti-social psychiatric disorders, which may have adverse effects on themselves and society as a whole [21,23]. If children under the age of five develop emotional/behavioral problems early in their growth, the risk of developing symptoms of a corresponding mental illness is high [24,25]. Therefore, it is necessary to find early signs of emotional/behavioral problems and identify high-risk groups for further diagnosis and treatment to prevent long-term effects of the diseases on a child’s mental health.

There are many ways to screen emotional and behavioral problems in children, of which the most commonly used are the parent’s Child Behavior Checklist (CBCL) and the Strength and Difficulties Questionnaire (SDQ) [26]. Compared to the CBCL, SDQ entries are simple, easy to administer, widely used, and freely available. SDQ is suitable for non-mental health professional investigators and respondents with a poor education level, which is conducive to the discovery of mental health problems in early childhood [26,27]. Various versions of the SDQ scale have been translated into Chinese and have good psychometric characteristics and reliability [27,28,29].

Research on LBC in China has mainly focused on the mental health of school-aged children [12,13,14,15,16], growth and development [30], unintentional injury [31], and health-seeking behaviors [32]. There is limited research available to explore the mental health status of preschool-aged LBC in rural China. We hypothesized that mental health problems are more common among rural LBC and the most important risks are their family environmental factors. Therefore, this study aimed to examine the prevalence of mental health issues among preschool-aged LBC by using SDQ and to determine the socio-demographic factors that influence the LBC’s emotional and behavioral problems. The results of this study may provide information for future mental health intervention for LBC.

## 2. Materials and Methods

The data for this study were derived from the baseline data of “The impact of conditional cash transfer on the nutritional status and physical development of 3–5 years old LBC in poor rural areas of China” [33].

### 2.1. Sampling

Hunan Province is located in the south-central part of China and more than half of the children in rural Hunan are LBC. Hunan is also an important minority province in China and the minorities in Western Hunan account for 75% of Hunan’s total minority population. Therefore, we selected Fenghuang County of Xiangxi Miao and Tujia Autonomous Prefecture as one research site, which is located in an impoverished mountainous area (Western Hunan). As a typical minority area, the minority population accounts for 78% in Fenghuang County, most of whom are Miao and Tujia minorities. We chose Pingjiang County of Yueyang City as another research site, which is located in a hilly and plain area (Eastern Hunan). The Han population in Pingjiang County accounts for 90% of the population. The geographical characteristics of the two counties are obviously different, which better represents the different geographies of Hunan Province [33,34]. Both counties have been nationally designated as key poverty-stricken counties and densely populated by LBC [33].

We used the random cluster sampling method. Of the 132 villages of Fenghuang County and 72 villages of Pingjiang County, Hunan Province, 20 villages were randomly selected from each county, and 15 LBC were randomly selected from each village.

The village inclusion criteria were as follows: Villages were included if they had a minimum of 15 LBC (aged 3–5 years old) living in poor households, defined as having an annual income <2300 Renminbi (RMB, Chinese currency, 1 RMB = 0.140 USD), and had no kindergarten or care centres for LBC. Villages were excluded if they received similar funding or benefits from other sources, such as charities or non-governmental organizations (NGOs).

The LBC inclusion criteria were as follows: Households caring for at least one LBC (3–5 years old), live in poor households, and defined as an average per-capita annual income lower than 2300 RMB. The LBC were excluded if households were receiving benefits from other charities, NGOs, or other similar programs.

### 2.2. Recruitment

With the assistance of local health workers, the research team identified those who met the inclusion criteria. Before the start of the study, caregivers of left-behind children (CLBC) who could participate in the study were provided with an explanation of the purpose, significance, and steps of the study in a language they could understand, as well as the possible risks and benefits of participating in the study, in order to obtain their informed consent. All caregivers were free to ask questions following the explanation of the study or to quit the research at any time. If they met the inclusion criteria, CLBC who signed the informed consent form were able to participate in the study.

### 2.3. Ethical Approval

This research was approved by the independent ethics committee of the Institute of Clinical Pharmacology, Central South University (registered number: ctxy-140003), and registered in the China Clinical Trial Register (registered number: ChiCTR-TRC-14005117). Informed consent was obtained from caregivers and all information was kept strictly confidential.

### 2.4. Data Collection

During the period from January to March 2015, we conducted a baseline survey to collect data on the basics of the LBC family and LBC’s emotional and behavioral problems.

Investigators were teachers and postgraduate students from Central South University. Each investigator was trained by psychologists and public health researchers from the university. All the questionnaires had been tested in our preliminary study and proved to have a good validity and reliability. Face-to-face interviews were used to guide the LBC’s main caregivers to fill out the relevant questionnaires. All items were explained to caregivers in a local language they could understand. During questionnaire interviews, quality control and guidance personnel were present to conduct validity checks and ensure the accuracy of the questionnaires.

### 2.5. Measures

CLBC were asked to indicate their child’s gender, age, ethnicity, and left-behind situation, the number of LBC aged 3 to 5 years at home, as well as their own gender, age, educational level, occupation, type of caregiver, willingness to care for LBC, burden of taking care of LBC, sick people at home, etc. Then, they were asked to complete the China parent-rated form of the SDQ extended version.

The Strength and Difficulties Questionnaire (SDQ) was compiled by American psychologist Goodman in 1997 to assess the emotional and behavioral problems of children and adolescents [35]. It was revised again in 2001 and has been introduced and applied in 60 countries and regions. It has good reliability and validity and is suitable for the evaluation and screening of emotional and behavioral problems among children and adolescents aged 3–16 years [36]. The SDQ contains 25 items, which includes 5 subscales: Emotional symptoms, conduct problems, hyperactivity-inattention, peer relationship problems, and prosocial behaviors. A three-point Likert-type scale was used to indicate the extent of a symptom: “Not true”, “Somewhat true”, or “Certainly true”. Each subscale consists of 5 items and scores range from 0 to 10 points. The sum of the first 4 subscales generates a Total Difficulties Score (ranging from 0 to 40 points) [35]. For the Total Difficulties Score and the first 4 subscales, a higher score indicates a greater likelihood of significant problems, while for the prosocial behavior subscale, a higher score corresponds to fewer problems. According to the norm set by the Shanghai Mental Health Center of China [37], the Total Difficulties Score was categorized as normal (0–13 points), borderline (14–16 points), and abnormal (17–40 points). Additionally, the other subscales were categorized as emotional symptoms (normal, 0–3 points; borderline, 4 points; abnormal, 5–10 points), conduct problems (normal, 0–2 points; borderline, 3 points; abnormal, 4–10 points), peer problems (normal, 0–2 points; borderline, 3 points; abnormal, 4–10 points), hyperactivity/inattention (normal, 0–5 points; borderline, 6 points; abnormal, 7–10 points), and prosocial behavior (normal, 10–6 points; borderline, 5 points; abnormal, 4–0 points).

### 2.6. Statistical Analysis

EpiData 3.0 software (The EpiData Association, Odense, Denmark) was used for data entry and the IBM SPSS 24.0 software package (IBM Corp., Armonk, NY, United Sates) was used for data analysis. The statistical methods used in this research include descriptive statistics, chi-squared tests, and multivariate logistic regression. Descriptive data were reported in the form of a percentage and *p* ≤ 0.05 was considered to be statistically significant. In this study, the primary outcome variable (SDQ behavioral problems) was defined as the abnormal cut-off of the Total Difficulties score (17–40 points), which was previously stated [37]. Multivariate logistic regression was performed to determine the socio-economic factors associated with LBC SDQ behavioral problems. 

## 3. Results

### 3.1. Sociodemographic Characteristics of LBC

The basic demographic characteristics of LBC are shown in Table 1. There were 557 LBC in this sample, including 296 boys and 261 girls. Of the LBC, 62.3% have a Han nationality, 75.3% of the parents of the LBC have gone out to work, 73.2% of the LBC have siblings, and 18.9% of the LBC have a sick person in their home. A total of 89.0% of caregivers were over 40 years old and most of them were women (60.0%). In total, 79.5% of the caregivers were below the primary school education level, 84.7% of the caregivers were grandparents of LBC, 96.8% of the caregivers were willing to care for LBC, and 78.8% of the caregivers felt the burden of care.

### 3.2. SDQ Banded Scores for LBC

Figure 1 shows the score of each subscale of the SDQ scale of three- to five-year-old LBC. In this study, 50.6% of the children’s difficulty scores were borderline/abnormal. The most common problem of LBC was hyperactivity/inattention, with a rate of 33.6%. The abnormal detection rates of peer problems, emotional problems, conduct problems, and pre-social behavior problems were 28.5%, 26.0%, 25.7%, and 17.4%, respectively.

### 3.3. Scores of SDQ Scales for LBC of Different Genders and Ages

Table 2 compares the differences of the different dimensions in the SDQ subscales for different ages and genders. The results showed that, compared with LBC aged 5 years, LBC aged 3 years had more conduct problems and inattention to hyperactivity (*p* < 0.05), and LBC aged 4 years had more pre-social behavioral problems (*p* < 0.05). There was no difference between boys and girls in terms of conduct problems, peer problems, and pre-social behavior problems. In emotional problems and hyperactivity disorder, there was a difference between boys and girls. Compared with boys, girls had more emotional problems (*p* < 0.05) and fewer hyperactivity disorders (*p* < 0.01).

### 3.4. Multivariate Logistic Regression

The LBC’s age, gender, ethnicity, and left-behind status, the number of LBC in the family, the sick people at home, and the CLBC’s age, gender, education level, burden to take care of LBC, and willingness to take care of LBC were included in the logistic regression to analyze the factors related to the emotional and behavioral problems of LBC. After adjusting for the LBC’s gender, ethnicity, and left-behind status, the number of LBC in the family, and the CLBC’s relationship to the LBC, age, gender, education level, and burden to take care of LBC, the factors related to the emotional and behavioral problems of LBC were the LBC’s age, the number of sick people at home, and the CLBC’s willingness to take care of the LBC (see Table 3). With three-year-old LBC as the reference variable, five-year-old LBC were less likely to have emotional and behavioral problems (Odds ratio, OR = 0.527; 95%; Confidence interval, CI (0.330–0.842)). Compared with no sick people at home, two sick people at home had a higher incidence of emotional and behavioral problems (OR = 12.560, 95% CI (2.549–61.902)). CLBC’s willingness to care for LBC was used as a reference variable and their unwillingness to care for LBC was more likely to result in emotional and behavioral problems (OR = 9.981, 95% CI (3.172–31.411)).

## 4. Discussion

In this study, a cross-sectional survey was conducted on the emotional and behavioral problems of LBC aged 3 to 5 years in poor rural areas of Hunan Province using SDQ.

Our study found that 27.6% of LBC aged 3 to 5 had abnormal SDQ scores. This is significantly higher than the value in the study of Xiaoyun Chen and his colleagues, where the detection rate of an abnormal SDQ score of 8900 preschool children aged 3–6 years in China was 13.6% [38]. Compared with other SDQ studies, the behavioral problems in preschool-aged LBC of this study were also greater than for preschool children of Denmark (3.6%), Britain (6.9%), and Germany (7.8%) [39,40,41]. Compared with preschool non-LBC, LBC had a higher risk of emotional and behavioral problems. On the one hand, the absence of parents of LBC may have a negative impact on the mental health of the child. In the sample population of this study, 75.3% of the parents of LBC were out, which was significantly higher than that reported by the Chinese Women’s Federation in 2013, which gave a figure of 46.74% for all the parents of LBC in rural areas [42]. On the other hand, LBC live in mountainous areas with inconvenient transportation. Poverty is very common and the utilization of basic public health services and social and economic levels are very low [43,44]. A systematic review study has shown that children from socially and economically disadvantaged families are approximately two–three times more likely to have abnormal emotional/behavioral problems than children from socially and economically advantageous families [45]. In addition, the majority of LBC have older grandparents with lower education as their caregivers. Previous studies have found that children living with grandparents are at a higher risk of emotional and behavioral disorders than children living with both parents [46]. The risk is two to four times that of children living with both parents [5]. This means that public health services should focus on the mental health of LBC in poor and remote areas.

Among the LBC we studied, boys had more hyperactivity than girls, which is consistent with previous research findings [40,41,47,48,49]. However, it is interesting that preschool left-behind girls had more emotional problems than preschool left-behind boys, which is in stark contrast to the fact that some Western countries have no gender differences in the emotional problems of preschool children [40,41,47]. A study in China also found that preschool non-left-behind girls have more emotional problems than preschool non-left-behind boys [38]. One possible reason for this is, in addition to the physiological characteristics and personality traits of boys and girls of different genders, families use different parenting styles for boys and girls. In a Chinese family, girls are expected to be gentle and compliant, while boys are expected to be brave, self-confident, and strong. Some researchers have studied the adaptation behavior of 2- to 5-year-old children in South Tyrol in Italy [50]. They found that even if people lived in the same area, there was still an ethnic difference in the childrearing of Italian and Austrian/German Italian which affected the adaptive behavior of their children. Therefore, we speculate that differences in Chinese and Western childrearing cultures may cause differences in emotional problems among preschool girls. In traditional Chinese families, boys are often more cared for than girls, and families are more concerned about boys than girls, especially in remote and impoverished rural areas [51]. This kind of care and love is often based on neglecting or even depriving girls of emotional support and material needs [52]. It is difficult for girls to get enough love and care in the family, and thus they have more emotional problems [53].

Multivariate analysis using binary logistic regression found that the LBC who were 5 years old, with two sick people at their home and CLBC unwilling to take care of them, were more likely to have emotional and behavioral problems. The younger the LBC, the higher the risk of emotional and behavioral problems. Younger LBC are more likely to have emotional and behavioral problems, and the same results have been obtained in other similar studies [12,16]. This may be because the parents of older children have had a relatively longer time away from their children to go out to work. As time goes by, a child’s discomfort due to the absence of one or both parents is gradually weakened, and the dependence of a child on their parents is gradually reduced. Therefore, their emotional and behavioral problems are reduced, and the detection rate is reduced [54]. The higher the number of sick people at home, the higher the risk of emotional and behavioral disorders among LBC of a preschool age. Relatives who were seriously ill in bed would undoubtedly increase the care burden of caregivers and the medical burden of the whole family. In the case of family financial constraints, young caregivers need to take care of their families, but also do farm work. Older caregivers themselves are weak and sick [3]. Therefore, in those families with seriously ill family members, the care of children may be worse and the LBC’s emotional needs may be more easily ignored. Therefore, the risk of emotional and behavioral disorders is increased. The CLBC’s willingness to care of children affected their emotional and behavioral development. Those who were unwilling to care for LBC increased the risk of the child’s emotional and behavioral problems. This is related to the willingness of guardians to influence the daily life and education of LBC. When guardians are unwilling to take care of LBC, the effectiveness of their guardianship is affected. LBC live in an environment where they are unwilling to be guarded and it is difficult to get proper care and warmth. When LBC encounter psychological confusion, it is difficult to obtain effective guidance and help, which will inevitably cause emotional fluctuations and the emotional behavior of LBC will be more prominent [54], leading to a borderline personality disorder and repeated suicide attempts [55,56]. 

In summary, solutions for mental health problems of LBC in poor rural areas in China are urgently needed. For LBC, family-based mental health interventions may not be effective. Most of the caregivers of LBC are older people with lower education levels and have a limited understanding of mental health. Although mobile phone parenting is encouraged for improving communication between parents and LBC, caution still needs to be exercised when explaining the effect of mobile phones on problems of family separation. Some rural areas have established “left-behind children homes”, but the facilities and staff have been quite different. Kindergartens and caregivers can be the best platform for mental health interventions for LBC. This addresses the challenges of a diverse family environment for LBC. Therefore, it is necessary to strengthen the grassroot construction in rural poverty-stricken areas, such as the establishment of kindergartens for pre-school children and a professional early childhood education team. At the same time, the rural traffic environment should be improved to promote the utilization of health services by rural LBC. For those remote rural areas, the Chinese government should strive to develop local economic industrial chains to provide adequate employment opportunities for adults to encourage young parents to return to their families.

This study is the first to report on the emotional and behavioral problems of LBC aged 3 to 5 years in poor rural areas of China. We chose face-to-face interviews to conduct SDQ questionnaire surveys of rural caregivers and received high responses. The Chinese SDQ has been proved to have a good validity and reliability, which can help determine the risk of early childhood mental health. This research will improve the awareness of primary health workers and caregivers regarding mental health for children and provide evidence for future interventions. The study has the following limitations. First, we used a cross-sectional study approach that cannot lead to a causal relationship between parental outings and these outcomes. Secondly, the information on emotional and behavioral problems of LBC originated from their caregivers, which may have led to information bias. Thirdly, although our sample can better represent the geographical characteristics of poverty-stricken rural areas in Hunan Province, it is still uncertain whether the research results can be extended to all LBC in Hunan Province or other poverty-stricken areas in China. Even though these are significant results, the interpretation is limited by wide confidence intervals. Future research should consider a larger sample size to better understand the relationships that have been shown to be significant in this initial study.

## 5. Conclusions

The detection prevalence rate of emotional and behavioral problems of three- to five-year-old LBC in poor rural areas was higher than that of children of the same age in urban areas and Western developed countries. There are gender differences in hyperactivity and emotional symptoms. Poor care increased the risk of emotional and behavioral abnormalities in the children. Interventions to give support to both LBC and their caregivers are urgently needed to promote LBC’s mental health and development.

## Figures and Tables

**Figure 1 ijerph-16-04188-f001:**
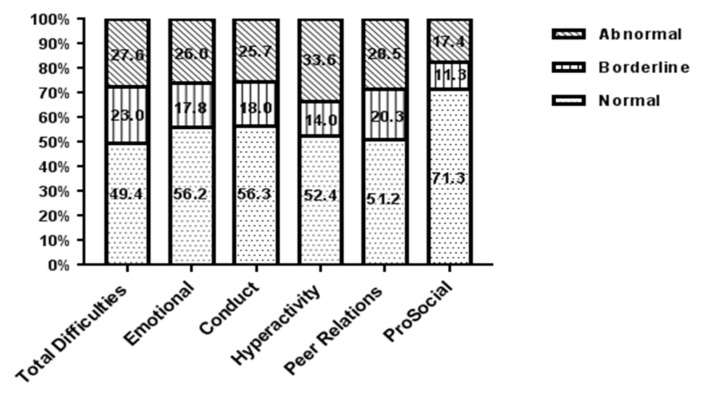
Strength and Difficulties Questionnaire (SDQ) banded scores for Hunan three- to five-year-old LBC in 2015.

**Table 1 ijerph-16-04188-t001:** Sociodemographic characteristics of the study population (n = 557).

Characteristics	Total Sample n (%)
**LBC’s age (years)**	
3	175 (31.4)
4	178 (32.0)
5	204 (36.6)
**LBC’s gender**	
Boy	296 (53.1)
Girl	261 (46.9)
**LBC’s ethnicity**	
Han	347 (62.3)
Minorities	210 (37.7)
**Left-behind status**	
Father out	111 (19.9)
Mother out	27 (4.8)
Both parents out	419 (75.3)
**Number of LBC in the family (n, %)**	
1	150 (26.8)
2	253 (45.6)
3 and above	154 (27.6)
**CLBC’s age (years)**	
20–39	61 (11.0)
40–59	261 (46.8)
60+	235 (42.2)
**CLBC’s gender**	
Male	223 (40.0)
Female	334 (60.0)
**CLBC’s education level**	
No formal education	172 (30.9)
Primary school	271 (48.6)
Middle school	93 (16.7)
High school	21 (3.8)
**CLBC’s relationship to LBC**	
Father	23 (4.1)
Mother	51 (9.2)
Grandparent	472 (84.7)
Other	11 (2.0)
**Sick person at home**	
None	452 (81.1)
1 sick person	95 (17.1)
2 sick people	10 (1.8)
**CLBC** **’s willingness to take care of LBC**	
Willing	539 (96.8)
Unwilling	18 (3.2)
**CLBC felt burden to take care of LBC**	
Having a burden	439 (78.8)
No burden	118 (21.2)

LBC, left-behind children; CLBC, caregiver of left-behind children.

**Table 2 ijerph-16-04188-t002:** SDQ domain among rural LBC by age and gender.

SDQ Domains	Age			Gender	
	3y (n = 175)	4y (n = 178)	5y (n = 204)	Boy (n = 296)	Girl (n = 261)
**Emotional symptoms ***					
Normal	89 (50.9%)	101 (56.7%)	123 (60.3%)	175 (59.1%)	138 (52.9%)
Borderline	34 (19.4%)	33 (18.5%)	32 (15.7%)	57 (19.3%)	42 (16.1%)
Abnormal	52 (29.7%)	44 (24.8%)	49 (24.0%)	64 (21.6%)	81 (31.0%)
**Conduct problems ^a^**					
Normal	85 (48.6%)	106 (59.6%)	123 (60.3%)	172 (58.1%)	142 (54.4%)
Borderline	31 (17.7%)	32 (18.0%)	37 (18.1%)	45 (15.2%)	55 (21.1%)
Abnormal	59 (33.7%)	40 (22.4%)	44 (21.6%)	79 (26.7%)	64 (24.5%)
**Hyperactivity **^,a^**					
Normal	79 (45.1%)	98 (55.1%)	115 (56.4%)	136 (45.9%)	156 (59.8%)
Borderline	25 (14.3%)	20 (11.2%)	33 (16.2%)	51 (17.3%)	27 (10.3%)
Abnormal	71 (40.6%)	60 (33.7%)	56 (27.4%)	109 (36.8%)	78 (29.9%)
**Peer Problems**					
Normal	84 (48.0%)	83 (46.6%)	118 (57.8%)	155 (52.4%)	130 (49.8%)
Borderline	34 (19.4%)	43 (24.2%)	36 (17.6%)	62 (20.9%)	51 (19.5%)
Abnormal	57 (32.6%)	52 (29.2%)	50 (24.6%)	79 (26.7%)	80 (30.7%)
**Prosocial behavior ^b^**					
Normal	116 (66.3%)	124 (69.6%)	157 (77.0%)	208 (70.3%)	189 (72.4%)
Borderline	23 (13.1%)	17 (9.6%)	23 (11.2%)	36 (12.2%)	27 (10.3%)
Abnormal	36 (20.6%)	37 (20.8%)	24 (11.8%)	52 (17.5%)	45 (17.3%)

Chi-square analysis. a: Age-3y group, compared with age-5y group, *p* < 0.05; b: Age-4y group compared with age-5y group, *p* < 0.05; boy group compared with girl group, * *p* < 0.05, ** *p* < 0.01.

**Table 3 ijerph-16-04188-t003:** Logistic regression model for exploring factors associated with SDQ behavioral problems (n = 557).

Variables	Adjusted OR (95% CI)	*p*-value
LBC’s age (years)		
4 y vs. 3 y	0.623 (0.388–1.001)	0.051
5 y vs. 3 y	0.527 (0.330–0.842)	0.007
Sick person at home		
1 sick person vs. none	1.401 (0.854–2.298)	0.182
2 sick people vs. none	12.56 (2.549–61.902)	0.002
CLBC’s willingness to take care of LBC		
Unwilling vs. willing	9.981 (3.172–31.411)	<0.001

SDQ behavioral problems refers to the Total Difficulties Score being abnormal (17–40 points); OR, odds ratio; CI, confidence interval. Adjusted for LBC’s gender, ethnicity, and left-behind status; the number of LBC in the family; and CLBC’s relationship to LBC, age, gender, education level, and burden to take care of LBC.

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
