# Peer review of "Emotional and Behavioral Problems Among 3- to 5-Year-Olds Left-Behind Children in Poor Rural Areas of Hunan Province: A Cross-Sectional Study"

_ijerph, 2019, doi:10.3390/ijerph16214188_

Round 1

Reviewer 1 Report

Thank you for the opportunity to review your work. The topic is very pertinent and worthwhile. Overall, the manuscript was interesting. My main concern is with how the outcome variable was defined, more comments on this and additional comments below: 

Line 21 - remove the additional space after SDQ 

Line 23- change is to was.  Since the study has already been completed, past tense is appropriate 

Line 27- same comment as above 

Line 39 -add a , in between 68766 so it reads 68,766 

Line 65 -specify if this reference is specific to American % or global.  It seems as if it's referring to American/ USA % but not quite clear as written. 

Line 92- Provide context to what influencing factors are being referred to. "This study analyses influencing and provides..." 

General comment on methods: in the study purpose it mentions that the aim of the manuscript is to investigate LBPC, however, sections 2 and 3 refer to only LBC.  Please clarify if the current study uses data of LBC or LBPC.  Based on the tables it seems to be LBPC, but the written information does not correspond. 

Line 95- use of the word derived.  Rephrase or remove. 

Line 99- in reference to half of the children in the area are LBC.  Is this LBC of all ages of only LPBC. 

Line 101- site should be plural 

Line 102- references the minority population, however, to readers who are not familiar with minority populations in China this should be further explained. 

Line 106- Please revise the sentence to be more clear.  Unsure of what "the location of both counties have been identified by the government as national key poverty counties"

Line 110- separate the sentence to clearly separate the village criteria from the family criteria since the criteria are different. 

Line 126- clearly define what is meant by choose freely. Choose freely to leave the study? to participate in the study? 

Line 134- reword after the comma to state we conducted a baseline survey to collect  (instead of that collected)

Line 136- reword to state more clearly what was done.  Additionally, not certain what training is being referred to at the beginning of the sentence. 

Lines 138-139- add information if the questionnaires were checked for completeness only or if questionnaires were also checked to be completed without error, etc. 

Line 142- use the CLBC acronym instead of caregiver if this is what you are referring to. 

Lines 147-163- please add information to clearly describe how emotional and behavioural problems were coded.  As written is not entirely clear how this is a dichotomous response warranting logistic regression. There are a variety of scales used and those are worthwhile, but further explanation of the outcome variable is needed. 

Line 167 - change statistical descriptions to descriptive statistics 

Table 1- bold the variable names to make the variables easily differ from the response options. 

Table 1- remove y~ after the ages since (years is provided in the title of the variable name  

Table 1- make sure all variable names align (number of LBC in family and CLBC's relationship to LBC are farther over than other variable names) 

Line 182- try to rephrase second sentence to not start with a number 

Table 2- if possible, draw a line down the middle of the table to separate Age and Gender 

Table 2- Remove ~ after each age 

Lines 203-206 revise to state increased odds or more or less likely instead of using the term "risk" as it implies that relative risk was calculated 

Line 245- change have to had 

Lines 283-285 reword these two sentences to make more clear 

Line 287 - change did not to cannot.  Cross sectional studies can not lead to a causal relationship because they cannot assess time 

Line 295- change rate to percent, percent prevalence, or prevalence rate to not confuse this information with an actual rate. 

Line 297- remove will and change increase to increases. 

Author Response

Cover Letter to Reviewer

Dear Reviewer,

Thank you for the insightful comments that help us to improve the quality of the paper. We have revised the article according to your suggestions.

Below we detail the point-to-point response to the reviewer’s comments. And you will also clearly see the difference made to the revised manuscript.

Sincerely yours,

Qian Lin,

Reviewer 1’s Comments and Suggestions for Authors

Thank you for the opportunity to review your work. The topic is very pertinent and worthwhile. Overall, the manuscript was interesting. My main concern is with how the outcome variable was defined, more comments on this and additional comments below: 

Comment 1Line 21 - remove the additional space after SDQ 

Response:  Thank you. We have removed the additional space.  

Comment 2Line 23- change is to was.  Since the study has already been completed, past tense is appropriate 

Response: We have amended it according to your suggestions, see line 28.

Comment 3Line 27- same comment as above 

Response: We have amended it according to your suggestions, see line 34

Comment 4Line 39 -add a , in between 68766 so it reads 68,766 

Response: We have amended it as “68,766”, see line 46

Comment 5Line 65 -specify if this reference is specific to American % or global.  It seems as if it's referring to American/ USA % but not quite clear as written. 

Response: Appreciate for this suggestions. We have changed it to “A recent national report” to specify American %, see lines 73.

Comment 6Line 92- Provide context to what influencing factors are being referred to. "This study analyses influencing and provides..." 

Response: We have amended the sentence as “Therefore, this study aims to examine the emotional and behavioural problems of preschool-aged LBC and association with socio-demographic factors in poverty-stricken rural areas, and to provide targeted reference for improving the health status of this special population.” , see line 103-107

Comment 7General comment on methods: in the study purpose it mentions that the aim of the manuscript is to investigate LBPC, however, sections 2 and 3 refer to only LBC.  Please clarify if the current study uses data of LBC or LBPC.  Based on the tables it seems to be LBPC, but the written information does not correspond. 

Response: Thank you for pointing out this. In order to be consistent with the title of this article and our previous articles from this study, we use LBC while not LPBC. In the method section, we have clarified that the study subjects were left-behind children aged 3 to 5 years old.

Comment 8Line 95- use of the word derived.  Rephrase or remove. 

Response: Thank you. We have amended it, See line 109.

Comment 9Line 99- in reference to half of the children in the area are LBC.  Is this LBC of all ages of only LPBC. 

Response: It is of all ages.  

Comment 10Line 101- site should be plural 

Response: We have rewritten this section. See line 114-126.

Comment 11Line 102- references the minority population, however, to readers who are not familiar with minority populations in China this should be further explained. 

Response: Thank you for the suggestions. We have rewritten this paragraph.

“Hunan Province is located in the south-central part of China, and more than half of the children in rural Hunan are LBC. Hunan is also an important minority province in China, and the minorities in the western Hunan account for 75% of Hunan’s total minority population. Therefore, we selected Fenghuang County of Xiangxi Miao and Tujia Autonomous Prefecture as one research site, which is located in an impoverished mountainous area (western Hunan). As the typical minority areas, the minority population accounts for 78% in Fenghuang County, most of them are Miao and Tujia minorities. We chose Pingjiang County of Yueyang City as another research site, which is located in a hilly and plain area (eastern Hunan). The Han population in Pingjiang County accounts for 90%. The geographical characteristics of the two counties are obviously different, which better represents the different geographies of Hunan Province. Both counties have been nationally designated as key poverty-stricken counties, and densely populated by left-behind children.” See line 113-126.

Comment 12Line 106- Please revise the sentence to be more clear.  Unsure of what "the location of both counties have been identified by the government as national key poverty counties"

Response: We have changed it to “Both counties have been nationally designated as key poverty-stricken counties, and densely populated by left-behind children.” See line 124-126.

Comment 13Line 110- separate the sentence to clearly separate the village criteria from the family criteria since the criteria are different. 

Response: We agree with you and have separated the criteria sentences to two short paragraphs. See line 131-139.

Comment 14Line 126- clearly define what is meant by choose freely. Choose freely to leave the study? to participate in the study? 

Response: Thank you for pointing out this. We have changed it to “All the caregivers were free to ask questions following the explanation or quit the investigation at any time.” See line 145-146.

Comment 15Line 134- reword after the comma to state we conducted a baseline survey to collect  (instead of that collected)

Response: We have amended it. See line 154.

Comment 16Line 136- reword to state more clearly what was done.  Additionally, not certain what training is being referred to at the beginning of the sentence. 

Response: Thank you. We have amended the sentences as “Investigators were the teachers and postgraduate students from Central South University. Each investigator was trained by psychologists and public health researchers from the university. Face-to-face interviews were used to guide the LBC’s main caregivers to fill out the relevant questionnaires. During questionnaires interviewing, the quality control and guidance personnel were present to conduct validity checks and accuracy of the questionnaires.” See line 156-163.

Comment 17Lines 138-139- add information if the questionnaires were checked for completeness only or if questionnaires were also checked to be completed without error, etc.

Response: Same as above.

Comment 18Line 142- use the CLBC acronym instead of caregiver if this is what you are referring to. 

Response: We have amended it. See line 166.

Comment 19Lines 147-163- please add information to clearly describe how emotional and behavioural problems were coded.  As written is not entirely clear how this is a dichotomous response warranting logistic regression. There are a variety of scales used and those are worthwhile, but further explanation of the outcome variable is needed. 

Response: Thank you for the comments. We have rewritten this part. “The SDQ includes 25 items, which are divided between 5 subscales: emotional symptoms, conduct problems, hyperactivity-inattention, peer relationship problems and prosocial behaviours. A three-point Likert-type scale was used to indicate extent a symptom, “Not true”, “Somewhat true”, or “Certainly true”. Each subscale consists of five items and scores range from 0-10 points. The sum of the first four subscales generates a total difficulties score (range from 0 to 40 points) [26]. For total difficulties score and the first four subscales, a higher score indicates a greater likelihood of significant problems; while for prosocial behavior subscale, a higher score correspond to fewer problems.” See line 175-190.

We also added two sentences to explain the primary outcome variable in 2.6. Statistical Analysis. “In this study, primary outcome variable (SDQ behavioural problems) was defined as abnormal cut-off of the total difficulties SDQ score (17-40 points), which previously stated. Multivariate logistic regression was performed to determine the socio-economic factors associated with LBC SDQ behavioural problems.” See line 202-205.  

Comment 20Line 167 - change statistical descriptions to descriptive statistics 

Response: Thank you. We have changed it to “descriptive statistics”. See line 200.

Comment 21Table 1- bold the variable names to make the variables easily differ from the response options. 

Response: Thanks for the suggestions. We have bolded the variable names.

Comment 22Table 1- remove y~ after the ages since (years is provided in the title of the variable name  

Response: We have amended it.

Comment 23Table 1- make sure all variable names align (number of LBC in family and CLBC's relationship to LBC are farther over than other variable names) 

Response: Thank you. We have amended it.

Comment 24Line 182- try to rephrase second sentence to not start with a number 

Response: We have added “In this study” as the start of the sentence

Comment 25Table 2- if possible, draw a line down the middle of the table to separate Age and Gender 

Response: Yes, we have added a line.

Comment 26Table 2- Remove ~ after each age 

Response: We have removed it and added “y”, see Table 2.

Comment 27Lines 203-206 revise to state increased odds or more or less likely instead of using the term "risk" as it implies that relative risk was calculated 

Response: Thank you for the comments. We have used “were less likely to have” to replace “had a lower risk of” to, and “were more likely to have” to replace “had a higher risk of”.

See line 246, 248 and 253.

Comment 28Line 245- change have to had 

Response: Amended.  

Comment 29Lines 283-285 reword these two sentences to make more clear 

Response: Thank you. We have rewritten these sentences as “We chose face-to-face interviews to conduct SDQ questionnaire surveys for the rural caregivers and received high responses. The Chinese SDQ has been proved to have good validity and reliability, which can help determine the risk of early childhood mental health. This research will improve the awareness of primary health workers and caregivers about children's mental health and provide evidences for future interventions.”  See lines 346-350

Comment 30Line 287 - change did not to cannot.  Cross sectional studies can not lead to a causal relationship because they cannot assess time 

Response: Thank you. We have amended it.

Comment 31Line 295- change rate to percent, percent prevalence, or prevalence rate to not confuse this information with an actual rate. 

Response: Thank you. We have amended it as “prevalence rate”.

Comment 32Line 297- remove will and change increase to increases. 

Response: Thank you. We have amended it.

Reviewer 2 Report

Thank you for inviting me to review the paper on “Emotional and behaviour problems among 3- to 5- year-old left-behind children in the poor rural areas of Hunan Province: a cross-sectional study”. The phenomenon of left-behind children is a unique problem in China and causes mental health problems at epidemiological level. This paper is of sound methodology and deserves to be published. I have the following recommendations:

The IJERPH is a leading journal on left-behind children in China. In order to improve the readers’ understanding of this problem, I recommend the authors to have a summary of the following articles in the introduction and mention the additional research question or value added by the current study:

References:

Wang F, Lin L, Xu M, et al Mental Health among Left-Behind Children in Rural China in Relation to Parent-Child Communication. Int J Environ Res Public Health. 2019 May 26;16(10). pii: E1855. doi: 10.3390/ijerph16101855. PMID: 31130670

Ma S, Jiang M, Wang F et al  Left-Behind Children and Risk of Unintentional Injury in Rural China-A Cross-Sectional Survey. Int J Environ Res Public Health. 2019 Jan 31;16(3). pii: E403. doi: 10.3390/ijerph16030403. PMID: 30708979

Ouyang Y, Zou J, Ji M et al  Study on the Status of Health Service Utilization among 3⁻5 Years Old Left-Behind Children in Poor Rural Areas of Hunan Province, China: A Cross-Sectional Survey. Int J Environ Res Public Health. 2019 Jan 4;16(1). pii: E125.

Zhou M, Sun X, Huang L et al  Parental Migration and Left-Behind Children's Depressive Symptoms: Estimation Based on a Nationally-Representative Panel Dataset. Int J Environ Res Public Health. 2018 May 24;15(6).

Guan H, Wang H, Huang J et al Health Seeking Behavior among Rural Left-Behind Children: Evidence from Shaanxi and Gansu Provinces in China.

Int J Environ Res Public Health. 2018 Apr 28;15(5). pii: E883. doi: 10.3390/ijerph15050883.

Hu H, Gao J, Jiang H et al  A Comparative Study of Behavior Problems among Left-Behind Children, Migrant Children and Local Children. Int J Environ Res Public Health. 2018 Apr 1;15(4). pii: E655. PMID: 29614783

Tian X, Ding C, Shen C et al  Does Parental Migration Have Negative Impact on the Growth of Left-Behind Children?-New Evidence from Longitudinal Data in Rural China.

Int J Environ Res Public Health. 2017 Oct 27;14(11). pii: E1308. doi: 10.3390/ijerph14111308.

PMID:29077043

Ji M, Zhang Y, Zou J et al  Study on the Status of Health Service Utilization among Caregivers of Left-Behind Children in Poor Rural Areas of Hunan Province: A Baseline Survey. Int J Environ Res Public Health. 2017 Aug 12;14(8). pii: E910. doi: 10.3390/ijerph14080910.

PMID: 28805702

This paper has very sound methodology. I hope the authors could mention the long term psychological implications (e.g. suicide and borderline personality disorder). Please modify the statement Pg 8 Line 278 – 280.

When LBC encounter psychological confusion, it is difficult to obtain effective guidance and help, which will inevitably cause emotional fluctuations, and the emotional behaviour of LBC will be more prominent [45], leading to borderline personality disorder (Keng et 2019) and repeated suicide attempts (Choo et al 2014).

References

Keng SL, Lee Y, Drabu S et al  Construct Validity of the McLean Screening Instrument for Borderline Personality Disorder in Two Singaporean Samples.

J Pers Disord. 2019 Aug;33(4):450-469

Choo C, Diederich J, Song I et al Cluster analysis reveals risk factors for repeated suicide attempts in a multi-ethnic Asian population.

Asian J Psychiatr. 2014 Apr;8:38-42

Author Response

Cover Letter to Reviewer

Dear Reviewer,

Thank you for the insightful comments that help us to improve the quality of the paper. We have revised the article according to your suggestions.

Below we detail the point-to-point response to the reviewer’s comments. And you will also clearly see the difference made to the revised manuscript.

Sincerely yours,

Qian Lin,

Thank you for inviting me to review the paper on “Emotional and behaviour problems among 3- to 5- year-old left-behind children in the poor rural areas of Hunan Province: a cross-sectional study”. The phenomenon of left-behind children is a unique problem in China and causes mental health problems at epidemiological level. This paper is of sound methodology and deserves to be published. I have the following recommendations:

Comment 1: The IJERPH is a leading journal on left-behind children in China. In order to improve the readers’ understanding of this problem, I recommend the authors to have a summary of the following articles in the introduction and mention the additional research question or value added by the current study:

References lists………….:

Response: Many thanks for your suggestions. We have added these references and quoted in background as reference 2, 3, 13, 14, 15, 30, 31 and 32.

Comment 2: This paper has very sound methodology. I hope the authors could mention the long term psychological implications (e.g. suicide and borderline personality disorder). Please modify the statement Pg 8 Line 278 – 280.

When LBC encounter psychological confusion, it is difficult to obtain effective guidance and help, which will inevitably cause emotional fluctuations, and the emotional behaviour of LBC will be more prominent [45], leading to borderline personality disorder (Keng et 2019) and repeated suicide attempts (Choo et al 2014).

Response: Deeply appreciate for the comments. We have revised these sentences according to your suggestions, and quoted the two references (reference 55 and 56). See line 325-328  

Reviewer 3 Report

Dear Authors,
I consider the data presented in your study to be very alarming about the children's health and well-being in the preschool age you have involved in your research.

For this reason I think it is important that you improve your study, so as to allow the scientific community to fully understand the health conditions of these children, the variables that influence their mental well-being, and the possible long-term consequences for their development.

I therefore ask you to respond to the methodological questions I raise, and to provide possible suggestions for social policies aimed at solving this very serious condition.
Finally, some small remarks concerning the style of your paper.

Best regards and good work.

I do not know if the authors tested the multicollinearity between the independent variables to ascertain if there is a problem with high correlation. For example, it is highly probable that low income families also have a low education. High correlation between independent variable could lead to problems in understanding which independent variable contributes to explain variance in the dependent variable.

Authors should comment more extensively the differences found between children's different age bands and different gender, considering the cultural context of the research and in comparison with findings of other studies in western countries. The importance of the findings of this study should be extensively explained to reader from other countries, that could find difficult to understand the characteristics of this population.

With respect to method, do researcher check the caregiver's ability to understand what was asked by the items of the questionnaires? People with a very low education could have difficulty to understand what is meant by the research instruments. Do they read alone the SDQs and other research instruments (Lines 142-146)? Were they able to read (30% did not have a formal education, and more than 40% had only primary education)?

I would expect the authors to offer some suggestions that would allow us to understand how to tackle such a complex and alarming problem as that of such a high incidence of emotional and behavioral problems in such young children (parent training?). Can parenting knowledge affect aspects related to the development and understanding of child development? See in this regard the discussion of the relationship between culture and parent knowledge in Taverna, L., Tremolada, M., Bonichini, S. (2017). Conoscenze materne e sviluppo del bambino in due gruppi culturali altoatesini. Ricerche di Psicologia, 40(2), 257-278. ISSN: 0391-6081. ISSNe: 1972-5620. DOI: 10.3280/RIP2017-002005.

Line 22: I would separate 27.6%/50.6% like this: 27.6%-50.6% Lines 106-107: please insert reference Lines 123-126: reformulate. Poor English Line 132: information were kept Lines 134-135: survey collecting data Lines 153-154: certainly true. Five items (7,....) are reverse coded. Line 174: patients? Table 2: bold the 5 years of age; gender boy/girl Line 192. inattention/hyperactivity lines 193-194: pro-social behaviour Lines 195-196: p, not capital letter Line 200: Multinomiallogistic regression? In this case the dependent variables are the scales of SDQ (emotional problems, behaviour problems and prosocial behavior with their three categories normal/abnormal/borderline). Please clarify. Lines 227-229: this sentence is not clear: "Compared with...Germany (7.8%)". Lines 251-252: another possible reason could be attributed to family culture. See for example the influence of family culture on child development when cohabiting cultures are studied (Taverna, L., Bornstein, M.H., Putnich, D.L., Axia, G. 2011. Adaptive Behaviors in young Children: A Unique Cultural Comparison in Italy. Journal of Cross Cultural Psychology, 42(3), 445–465. DOI: 10.1177/0022022110362748).

Author Response

Cover Letter to Reviewer

Dear Reviewer,

Thank you for the insightful comments that help us to improve the quality of the paper. We have revised the article according to your suggestions.

Below we detail the point-to-point response to the reviewer’s comments. And you will also clearly see the difference made to the revised manuscript.

Sincerely yours,

Qian Lin,

Dear Authors,

I consider the data presented in your study to be very alarming about the children's health and well-being in the preschool age you have involved in your research.

For this reason I think it is important that you improve your study, so as to allow the scientific community to fully understand the health conditions of these children, the variables that influence their mental well-being, and the possible long-term consequences for their development.

I therefore ask you to respond to the methodological questions I raise, and to provide possible suggestions for social policies aimed at solving this very serious condition.

Finally, some small remarks concerning the style of your paper.

Best regards and good work.

Comment 1: I do not know if the authors tested the multicollinearity between the independent variables to ascertain if there is a problem with high correlation. For example, it is highly probable that low income families also have a low education. High correlation between independent variable could lead to problems in understanding which independent variable contributes to explain variance in the dependent variable.

ResponseMany thanks for your comments. We understand reviewer’s concern. We did the collinearity diagnostics and the results haven’t show significant multicollinearity between the associate factors.

Comment 2: Authors should comment more extensively the differences found between children's different age bands and different gender, considering the cultural context of the research and in comparison with findings of other studies in western countries. The importance of the findings of this study should be extensively explained to reader from other countries, that could find difficult to understand the characteristics of this population.

ResponseThank you. We have added several sentences “………., families use different parenting styles for boys and girls. In a Chinese family, girls are expected to be gentle, and compliant, while boys are expected to be brave, self-confident, and strong. Some researchers have studied the adaptation behavior of 2-5 year old children in South Tyrol in Italy[50]. They found that even if the people lived in the same area, there is still an ethnic difference in childrearing of Italian and Austrian/German Italian which affect the adaptive behavior of their children. Thus, we speculate that differences in Chinese and Western childrearing cultures may cause differences in emotional problems among preschool girls.”  See line 292-298

Comment 3: With respect to method, do researcher check the caregiver's ability to understand what was asked by the items of the questionnaires? People with a very low education could have difficulty to understand what is meant by the research instruments. Do they read alone the SDQs and other research instruments (Lines 142-146)? Were they able to read (30% did not have a formal education, and more than 40% had only primary education)?

Response: Thank you for pointing out this. We have rewritten the section of “Data collection” to describe how questionnaires been investigated. Investigators were the teachers and postgraduate students from Central South University. Each investigator was trained by psychologists and public health researchers from the university. All the questionnaires were tested in our preliminary study and have been proved to have good validity and reliability. Face-to-face interviews were used to guide the LBC’s main caregivers to fill out the relevant questionnaires. All the items were explained to caregivers in the local language that they could understand. During questionnaires interviewing, the quality control and guidance personnel were present to conduct validity checks and accuracy of the questionnaires.”  See line 156-164.

Comment 4: I would expect the authors to offer some suggestions that would allow us to understand how to tackle such a complex and alarming problem as that of such a high incidence of emotional and behavioral problems in such young children (parent training?). Can parenting knowledge affect aspects related to the development and understanding of child development? See in this regard the discussion of the relationship between culture and parent knowledge in Taverna, L., Tremolada, M., Bonichini, S. (2017). Conoscenze materne e sviluppo del bambino in due gruppi culturali altoatesini. Ricerche di Psicologia, 40(2), 257-278. ISSN: 0391-6081. ISSNe: 1972-5620. DOI: 10.3280/RIP2017-002005.

Response: Thank you for the suggestion. We have added one paragraph to offer some suggestions.

“In summary, the solution for mental health problems of left-behind children in poor rural areas in China are urgently needed. For left-behind children, family-based mental health interventions may not be effective. Most of the caregivers of left-behind children are older people with lower education levels and have limited understanding of mental health. Although mobile phone parenting is encouraged for improving communication between parents and LBC, it still needs to be cautious when explain the effect of mobile phone on problems of family separation. Some rural areas have established "left-behind children homes", but the facilities and staff were quite different. Kindergartens and caregivers can be the best platform for mental health interventions for left-behind children. It addresses the challenges of a diverse family environment for left-behind children. Therefore, it is necessary to strengthen the grassroots construction in rural poverty-stricken areas, such as the establishment of kindergartens for pre-school children, and a professional early childhood education team. At the same time, the rural traffic environment should be improved to promote the health services utilization for the rural LBC. For those remote rural areas, the Chinese government should strive to develop local economic industrial chains to provide adequate employment opportunities for adults to encourage the young parents back to their families.”  See line 329-343

Comment 5: Line 22: I would separate 27.6%/50.6% like this: 27.6%-50.6% Lines 106-107:

Response: We have revised it.

Comment 6: please insert reference Lines 123-126: reformulate.

Response: Thank you. We have inserted the references.

Comment 6: Poor English Line 132: information were kept Lines 134-135: survey collecting data Lines 153-154: certainly true. Five items (7,....) are reverse coded. Line 174: patients? Table 2: bold the 5 years of age; gender boy/girl Line 192. inattention/hyperactivity lines 193-194: pro-social behaviour Lines 195-196: p, not capital letter

Response: Thank you. We have revised the text according to your and other reviewers’ comments. See line 152, line 154, line 179-182, line 211, Table 1 and Table 2. 

Comment 7: Line 200: Multinomiallogistic regression? In this case the dependent variables are the scales of SDQ (emotional problems, behaviour problems and prosocial behavior with their three categories normal/abnormal/borderline). Please clarify.

Response: Thank you. We have added a footnote to Table 3. “SDQ behavioural problems refers to the Total Difficulties Score was abnormal (17-40 points)”. We also defined primary outcome in 2.6 Statistical Analysis “In this study, primary outcome variable (SDQ behavioural problems) was defined as abnormal cut-off of the total difficulties SDQ score (17-40 points), which previously stated. Multivariate logistic regression was performed to determine the socio-economic factors associated with LBC SDQ behavioural problems.”  See line 202-205.

Comment 7:Lines 227-229: this sentence is not clear: "Compared with...Germany (7.8%)".

Response: We have amended it as “Compared with other SDQ studies, the behavioural problems in preschool aged LBC of this study were also higher than the preschool children of Denmark (3.6%), Britain (6.9%) and Germany (7.8%) ”  See line 266-270.

Comment 8: Lines 251-252: another possible reason could be attributed to family culture. See for example the influence of family culture on child development when cohabiting cultures are studied (Taverna, L., Bornstein, M.H., Putnich, D.L., Axia, G. 2011. Adaptive Behaviors in young Children: A Unique Cultural Comparison in Italy. Journal of Cross Cultural Psychology, 42(3), 445–465. DOI: 10.1177/0022022110362748).

Response: Appreciate for the suggestion. We have added some sentences and quoted the reference. “Another possible reason could be attributed to family culture. Some researchers have studied the adaptation behavior of 2-5 year old children in South Tyrol in Italy[50]. They found that even if the people lived in the same area, there is still an ethnic difference in childrearing of Italian and Austrian/German Italian which affect the adaptive behavior of their children. Thus, we speculate that differences in Chinese and Western childrearing cultures may cause differences in emotional problems among preschool girls.” See line 292-298.

Round 2

Reviewer 1 Report

Thank you for the revisions to the manuscript and the opportunity to review the manuscript again. 

Overall, the revisions strengthened the manuscript.  The manuscript could benefit from copy editing.  Additional comments below: 

Be consistent with use of left-behind children and LBC.  Once the acronym has been spelled out once the acronym can be used throughout the rest of the manuscript. 

Line 22: extra space between (SDQ) and the . 

Line 45: Change we also found to "the literature also indicates that" so the reader is not confused as to whether that was a finding from this study, previous research from this team, or from the literature.  It's cited as other articles, but the modification in language will assist the flow. 

Line 96: sentence doesn't not seem complete or is missing proper grammar. 

Line 110: Add "of the population" to the end of the sentence about the Han population in Pingjiang County. 

Line 113: change left-behind children to LBC. 

Line 131: add "of the study" after following the explanation. 

Line 132: change investigation to interview or research since participants weren't being investigated. 

Line 141: change their to LBC since the sentence is referring to LBC emotional and behavioural problems, not emotional and behavioural problems of the LBC family.

Line 142: 148 remove the italics. 

Line 144: reliability is misspelled

Line 148: Move "2.5 Measures" to the next line

Line 171: extra space after 7-10 points) 

Line 178-181: specify how the outcome variable was dichotomized. In the methods, there are multiple levels for SDQ scores.  It is inferred that the outcome variable is dichotomized into SDQ score 0-16 versus SDQ score 17-40, but that needs to be clear. 

Line 184: change are to were

Line 186: add the word "a" between have and sick 

Table 1: remove y after each age in LBC age; add (years) after CLBC's age to be consistent with how LBC's age was reported; remove y~ after ages in CLBC's age; clarify the CLBC's age ranges (20-39; 40-59; 60+) for example

Results: be consistent when reporting p values, some are capitalized some are lower case, some are italicized some are not. 

Line 223-Line 229: Be clear if results are significant when reporting them.  The only significant results are Age 5 versus Age 3; 2 sick people versus none; and CLBC willingness. 

Line 248: add the word "who" after LBC

Line 276-280: adjust to better reflect the significant results. As reported is seems like all variables and all response options within the variable were significant. 

Line 289-291: this sentence is confusing as written, try to reword. 

Line 299-300: extra comma and extra period 

Discussion: Add some information regarding the results that have very wide confidence intervals (2 sick people in the home; unwillingness to care for LBC).  These are significant results but the confidence intervals are quite wide due to the small sample size. It would be useful in future research to have a larger sample size to better understand these relationships that have been shown to be significant in this initial study. 

Author Response

Dear reviewer,

Deeply appreciate for your comments!

You have given us a lot of insightful and helpful suggestions for revising and improving our paper.  Many thanks! Here we submitted the round 2 cover letter and the manuscript. 

Best regards,

Reviewer 3 Report

Dear Authors,

I have read your paper again and I think you have improved it with careful changes.

Hope that this paper will find scientific resonance, since issue and data are very important for children wellbeing.

Best regards.

Author Response

Dear reviewer,

Deeply appreciate for your comments!

You have given us a lot of insightful and helpful suggestions for revising and improving our paper.  Many thanks!

Best regards,